Descriptive anatomy of the largest known specimen of Protoichthyosaurus prostaxalis (Reptilia: Ichthyosauria) including computed tomography and digital reconstruction of a three-dimensional skull

Lomax Dean R. 1 dean.lomax@manchester.ac.uk
http://orcid.org/0000-0002-0546-2381 Porro Laura B. 2
Larkin Nigel R. 3
1 School of Earth and Environmental Sciences, University of Manchester , Manchester , UK
2 Department of Cell and Developmental Biology, University College London , London , UK
3 Cambridge University Museum of Zoology , Cambridge , UK
Young Mark
Electronic publication date: 2019 Jan 8
Publication date: 2019
Volume: 7
Electronic Location ID: e6112
Received 2018 Aug 9; Accepted 2018 Nov 13
Copyright: © 2019 Lomax et al.
Copyright year: 2019
Copyright holder: Lomax et al.
License: This is an open access article distributed under the terms of the Creative Commons Attribution License, which permits unrestricted use, distribution, reproduction and adaptation in any medium and for any purpose provided that it is properly attributed. For attribution, the original author(s), title, publication source (PeerJ) and either DOI or URL of the article must be cited.
License URL: https://creativecommons.org/licenses/by/4.0/

Keywords: Ichthyosauria, Ichthyosauridae, Visualization, CT-scanning

Funding: PRISM fund, Arts Council, England The Dorothy and Edward Cadbury Trust The Curry Fund of the Geologists’ Association Birmingham Museums Trust Dean’s Doctoral Scholarship Award from the University of Manchester Funding for the conservation work, CT scanning, rebuilding of the skull and redisplay was received from: PRISM fund, Arts Council, England; The Dorothy and Edward Cadbury Trust; The Curry Fund of the Geologists’ Association; and internal funding from the Birmingham Museums Trust. DRL’s travel was covered in part by a PGR, Dean’s Doctoral Scholarship Award from the University of Manchester. The funders had no role in study design, data collection and analysis, decision to publish, or preparation of the manuscript.

==============================
Ichthyosaur fossils are abundant in Lower Jurassic sediments with nine genera found in the UK. In this paper, we describe the partial skeleton of a large ichthyosaur from the Lower Jurassic (lower Sinemurian) of Warwickshire, England, which was conserved and rearticulated to form the centrepiece of a new permanent gallery at the Thinktank, Birmingham Science Museum in 2015. The unusual three-dimensional preservation of the specimen permitted computed tomography (CT) scanning of individual braincase elements as well as the entire reassembled skull. This represents one of the first times that medical imaging and three-dimensional reconstruction methods have been applied to a large skull of a marine reptile. Data from these scans provide new anatomical information, such as the presence of branching vascular canals within the premaxilla and dentary, and an undescribed dorsal (quadrate) wing of the pterygoid hidden within matrix. Scanning also revealed areas of the skull that had been modelled in wood, clay and other materials after the specimen’s initial discovery, highlighting the utility of applying advanced imaging techniques to historical specimens. Additionally, the CT data served as the basis for a new three-dimensional reconstruction of the skull, in which minor damage was repaired and the preserved bones digitally rearticulated. Thus, for the first time a digital reconstruction of the skull and mandible of a large marine reptile skull is available. Museum records show the specimen was originally identified as an example of Ichthyosaurus communis but we identify this specimen as Protoichthyosaurus prostaxalis. The specimen features a skull nearly twice as long as any previously described specimen of P. prostaxalis, representing an individual with an estimated total body length between 3.2 and 4 m.

Introduction

Ichthyosaurs were a highly successful group of predatory marine reptiles that appeared in the late Early Triassic and went extinct in the early Late Cretaceous (Fischer et al., 2016). Some of the earliest forms were ‘lizard-like’ in appearance, although later forms evolved fish-shaped bodies (Motani, 2009). Species ranged in size from small-bodied forms less than one m long to giants over 20 m in length (Motani, 2005; Nicholls & Manabe, 2004; Lomax et al., 2018). Numerous Lower Jurassic ichthyosaurs have been found in the UK, the majority being from the Lyme Regis–Charmouth area in west Dorset (Milner & Walsh, 2010), the village of Street and surrounding areas in Somerset (Delair, 1969), sites around the coastal town of Whitby, Yorkshire (Benton & Taylor, 1984) and Barrow-upon-Soar, Leicestershire (Martin, Frey & Riess, 1986). Notable specimens have also been recorded from Ilminster, Somerset (Williams, Benton & Ross, 2015), Nottinghamshire (Lomax & Gibson, 2015) and Warwickshire (Smith & Radley, 2007), with various isolated occurrences at other sites across the UK (Benton & Spencer, 1995).

A partial ichthyosaur skeleton (BMT 1955.G35.1—Birmingham Museums Trust) was discovered in 1955 in Warwickshire, England. The specimen comprises a largely complete skull, portions of the pectoral girdle, pelvis, fore- and hindfins and numerous vertebrae and ribs. Bones of the basicranium and palate were also found, which are rarely observed in association with Lower Jurassic ichthyosaur skulls (Marek et al., 2015). The skull bones were reassembled three-dimensionally on a wood and metal frame held together with alvar, jute and kaolin dough, with missing parts carved from wood; however, some aspects were not accurately reconstructed. Museum records indicate that BMT 1955.G35.1, which has never been formally described, was originally identified as an example of Ichthyosaurus communis De la Beche & Conybeare 1821.

In 2015, as part of the development of the new Marine Worlds Gallery at the Thinktank, Birmingham Science Museum, the skull was dismantled, conserved and reassembled to be more anatomically accurate. The skull and postcranial skeleton of BMT 1955.G35.1 were publicly displayed for the first time, forming the centrepiece of this permanent gallery. The skull of BMT 1955.G35.1 is preserved in 3D and is free of matrix; this contrasts with the majority of Lower Jurassic ichthyosaur skulls, which are often flattened or displaced and preserved in matrix, enabling a more detailed description than is typical. The large size of many marine reptile skulls has precluded attempts to visualize specimens using medical imaging (but see McGowan, 1989). Given the exceptional 3D preservation, the fact it is relatively free of matrix, and access to facilities capable of imaging large specimens, we took the opportunity to scan individual cranial elements as well as the entire skull of BMT 1955.G35.1 using computed tomography (CT) before and after reassembly. CT and 3D digital reconstruction are increasingly being applied to the skulls of fossil vertebrates, including early tetrapods (Porro, Rayfield & Clack, 2015a, 2015b), dinosaurs (Rayfield et al., 2001; Lautenschlager et al., 2014, 2016; Porro, Witmer & Barrett, 2015c; Button, Barrett & Rayfield, 2016) and extinct synapsids (Wroe, 2007; Jasinoski, Rayfield & Chinsamy, 2009; Sharp, 2014; Cox, Rinderknecht & Blanco, 2015; Lautenschlager et al., 2017). The first attempt to understand the internal anatomy of the ichthyosaur skull was carried out by Sollas (1916) using serial grinding; although this method produced excellent understanding of skull anatomy, it was time-consuming, labour intensive and resulted in the destruction of the specimen. In contrast, modern medical imaging methods have been applied only to isolated regions of fossil marine reptile skulls (Kear, 2005; Fernández et al., 2011; Sato et al., 2011; Neenan & Scheyer, 2012; Herrera, Fernández & Gaspirini, 2013), with the exception of one pliosaur (Foffa et al., 2014a), one small ichthyosaur (Marek et al., 2015), for which entire skulls were CT scanned, and the skeleton of a juvenile plesiosaur (Larkin, O’Connor & Parsons, 2010).

In this paper, we use CT scanning of a large ichthyosaur skull along with careful examination of the original specimen to formally describe BMT 1955.G35.1. Based on this description we reassign the specimen to Protoichthyosaurus prostaxalis Appleby, 1979, a genus recently shown to be distinct from Ichthyosaurus based on multiple characters (Lomax, Massare & Mistry, 2017; Lomax & Massare, 2018). Furthermore, the studied specimen has an estimated maximum body length of 3.2–4 m, greater than any other known specimen of Protoichthyosaurus or Ichthyosaurus.

Geological Setting

BMT 1955.G35.1 was collected in situ from Fell Mill Farm, between Shipston-on-Stour and Honington, Warwickshire, England, grid reference NGR SP 277 415. The initial discovery was made by Mr Michael Bryan in May, 1955. A complete excavation, under the supervision of Assistant Keeper of Natural History at the City of Birmingham Museum, Mr Vincent Smith, subsequently took place. The specimen was found approximately four feet below the ground surface in a hard, blue-grey clay, lying directly on top of a brown grit layer containing numerous Gryphaea bivalves. Due to the fragmentary nature of the bones, they were removed embedded in clay.

Precise stratigraphic data associated with the discovery are not available but the remains were recorded as being from Liassic sediments, which conforms to the Early Jurassic age of the region’s geology (Edmonds, Poole & Wilson, 1965; Radley, 2003; Smith & Radley, 2007). In addition to the ichthyosaur skeleton, other fossils were collected alongside the specimen, including Gryphaea bivalves, a plesiosaur vertebra and an isolated shark tooth identified as Hybodus cf. H. cloacinus Quenstedt, 1858, which are also Early Jurassic in age, although this shark species ranges from the Rhaetian through Lower Lias (NRL, personal communication, D. Ward, 2015). Additionally, we found an ammonite fragment stored with the specimen, which is an example of Euagassiceras sauzeanum (d’Orbigny, 1844), a species indicative of the Semicostatum Ammonite Zone, lower Sinemurian, Lower Jurassic (DRL, personal communication, M. Howarth, 2017). As there was no record stating whether this ammonite fragment was physically collected with BMT 1955.G35.1, NRL was given permission by the current owners of Fell Mill Farm to collect other fossils along with matrix from the original site at a depth of two m below the surface. This resulted in the collection of numerous ammonites identified as Arnioceras semicostatum (Young & Bird, 1828), which is also indicative of the lower Sinemurian, Semicostatum Ammonite Zone (DRL, personal communication, M. Howarth, 2017). Thus, associated ammonites have provided the stratigraphic position of BMT 1955.G35.1.

Material and Methods

BMT 1955.G35.1 is currently housed in the Thinktank Science Museum (TSM). It was originally accessioned into the collections of Birmingham Museum and Art Gallery (BMAG) and loaned to TSM. However, BMAG and TSM have since become part of the BMT. The postcranial skeleton, long considered ‘missing’, was rediscovered in the collections of the Lapworth Museum of Geology (BU) and reunited with the skull as part of a funded project at the TSM. As BMT 1955.G35.1 was largely undeformed, the individual skull bones were assembled in 3D; however, several errors were made in this original reconstruction (Fig. 1A). As part of the funded project, the skull was disassembled and the individual bones cleaned, conserved and remounted (Figs. 1B and 1C). Many of the preserved skull bones were disarticulated when discovered and several cranial bones are not represented. The teeth have been reset and are not in their original positions. Portions of some elements are poorly preserved and/or taphonomically distorted, which somewhat restricts our description; for example, the dentaries cannot be articulated at the symphysis or mounted in their correct anatomical position. The newly reassembled skull of BMT 1955.G35.1 is based on all the preserved elements robust enough to safely include, and we limit our description of sutural contacts to those between elements preserved in original articulation. Specific details of the reconstruction and conservation of the studied specimen will be dealt with in a separate paper.

Figure 1 Three-dimensional skull of BMT 1955.G35.1, Protoichthyosaurus prostaxalis.

(A) Original photograph of the first skull reconstruction (left lateral view) within a couple of years of the 1955 excavation. Note that the prefrontal and postorbital are present, which we have been unable to locate in our study. (B) Skull in left lateral view, as reconstructed in 2015. (C) Skull in right lateral view, as reconstructed in 2015. Note the distinctive asymmetric maxilla with long, narrow anterior process. Teeth are not in their original positions. Scale bar represents 20 cm.

Prior to remounting, several individual bones of the left side of the skull were scanned using microcomputed tomography (μCT) in March 2015 at the Cambridge Biotomography Centre (Zoology Department, University of Cambridge) on an X-Tek H 225 μCT scanner (Nikon Metrology, Tring, UK) at 135 kV and 227 μA with no filtering. Elements scanned individually include: the left articular, opisthotic, stapes, quadrate and pterygoid; the median supraoccipital and basisphenoid; and both parietals. Voltage, current and resolution (0.1 mm/voxel) were identical for all scans. Scan data were visualized in the software Avizo 8.0 (Thermo Fisher Scientific, Waltham, MA, USA) and the left-side elements mirrored across the sagittal midline. All 3D surfaces were exported as stereolithography files and 3D printed at life-size in gypsum on a 3DS x60 3D Printer; pieces were subsequently dipped in cyanoacrylate for strength (NRL, personal communication, S. Dey, 2016).

After remounting, the skull of BMT 1955.G35.1, including the 3D printouts previously described, was scanned in May 2015 at the Royal Veterinary College on a Lightspeed Pro 16 CT scanner (GE Medical Systems Ltd., Pollards Wood, UK) at 120 kV and 200 μA. Due to the size of the specimen, it was scanned in two parts—the front of the skull was scanned at 0.56 × 0.56 × 1.25 mm/voxel and the rear of the skull was scanned at 0.73 × 0.73 × 1.25 mm/voxel. Both scans produced a total of 2,168 DICOM slices. Both scans used an exposure time of 2,356 ms and a body filter. Density thresholding was used to separate higher-density fossil bone from lower-density matrix as well as areas of the skull historically modelled in wood, clay and jute and portions newly 3D printed in gypsum. Scans were segmented to isolate individual bones and teeth, and to trace internal features. The two halves of the skull were overlain and merged using skeletal landmarks visible in both datasets (Figs. 2–4). Three-dimensional surfaces were exported as wavefront (OBJ) files to create an interactive 3D PDF using Tetra4D Reviewer and Converter (Tech Soft 3D, Bend, OR, USA) and Adobe Acrobat Pro X (Adobe Systems, San Jose, CA, USA). This reconstruction is provided as Supplemental Information (Appendix S1) and are the basis for the following description.

Figure 2 Surface models (generated from CT scan data) of preserved bones from the upper jaw of BMT 1955.G35.1, Protoichthyosaurus prostaxalis.

Right (A) and left (B) lateral views of the cranium. Medial views of the right (C) and left (D) sides of the cranium. Dorsal (E) and ventral (F) views of the cranium. Lateral views of the right (G) and left (H) premaxillae. Dorsal views of the right (I) and left (J) premaxillae. Posterior (K) view of the upper jaw. Individual bones are shown in different colours. Bones in (G)–(J) are transparent to visualize internal canals (shown in red opaque). Teeth are not in their original positions. Abbreviations: bs, basisphenoid; f?, possible fragment of frontal; j, jugal; l, lacrimal; mx, maxilla; n, nasal; op, opisthotic; p, parietal; pf, prefrontal; pmx, premaxilla; pt, pterygoid; q, quadrate; so, supraoccipital; sp, supratemporal; st, stapes. Scale bars equal 10 cm.

Figure 3 Surface models (generated from micro-CT scan data) of preserved palatal and braincase bones from BMT 1955.G35.1, Protoichthyosaurus prostaxalis.

Right medial (A) and left lateral (B) views, dorsal (C) and ventral (D) views and anterior (E) and posterior (F) views. Isolated supraoccipital in right anterolateral view (G). Individual bones are shown in different colours. Supraoccipital (G) is transparent to visualize internal canals (shown in red opaque). Abbreviations: bs, basisphenoid; f?, probable fragment of upper pterygoid wing; op, opisthotic; p, parietal; pt, pterygoid; q, quadrate; se, sella turcica; so, supraoccipital; sp, supratemporal; st, stapes. Scale bars equal 10 cm, except for (G) which equals five cm.

Figure 4 Surface models (generated from CT scan data) of preserved bones from the lower jaw of BMT 1955.G35.1, Protoichthyosaurus prostaxalis.

Lateral views of the right (A) and left (B) lower jaws. Medial views of the right (C) and left (D) lower jaws. Dorsal (E) and ventral (F) views of both halves of the lower jaws. Lateral views of the right (G) and left (H) dentaries. Ventral views of the right (I) and left (J) dentaries. Lateral oblique (K) view of the left surangular. Individual bones are shown in different colours. Bones in (G)–(K) are transparent to visualize internal canals (shown in red opaque). Teeth are not in their original positions. Abbreviations: an, angular; ar, articular, d, dentary; f?, possible surangular fragment; sa, surangular; sp, splenial; spf, splenial fragment. Scale bars equal 10 cm.

Surface models of individual bones were manipulated in 3D space using the Transform Editor within Avizo, allowing digital 3D reconstruction of the skull of BMT 1955.G35.1 following similar methods applied to early tetrapods (Porro, Rayfield & Clack, 2015a, 2015b) and dinosaurs (Lautenschlager, 2016). Most of the bones in the digital reconstruction are from the left side of BMT 1955.G35.1 as this side is generally better preserved. Minor damage was manually repaired in the Segmentation Editor within Avizo using interpolation, including: minor breaks and missing alveolar margins in the left premaxilla, maxilla, dentary and splenial; minor breaks in the left nasal, lacrimal, jugal, quadrate, pterygoid and parietal; the missing right margin of the supraoccipital; and gaps within the anterior half of the left surangular. Portions of bones preserved on the right but absent on the left—including the posterior tip of the right jugal and anterior tip of the right splenial—were duplicated, reflected across the sagittal midline and merged with left side elements using anatomical landmarks. We did not attempt to reconstruct missing bones or preserved elements that could not be scanned due to their delicate nature (see Results). The disarticulated bones were then fitted together at sutural contacts; we also referred to known relationships between skull bones from other ichthyosaur skulls (Andrew, 1910; Sollas, 1916; McGowan, 1973; Kirton, 1983; McGowan & Motani, 2003; Marek et al., 2015; Moon & Kirton, 2016). Lastly, left side elements were duplicated and reflected to form the right side of the skull. Transformation matrices for all bones from the original data set to the final 3D reconstruction are available as Supplemental Information (Appendix S2); a 3D PDF of the reconstructed skull is also available as Supplemental Information (Appendix S3).

Systematic Palaeontology

Ichthyosauria De Blainville, 1835

Parvipelvia Motani, 1999

Ichthyosauridae Bonaparte, 1841

Protoichthyosaurus Appleby, 1979

Protoichthyosaurus prostaxalis Appleby, 1979

Type species. Protoichthyosaurus prostaxalis Appleby, 1979. The type series of specimens are from historic collections. However, the holotype is most likely from the area around Street, Somerset and is most likely from the lowermost Jurassic (lower Hettangian) ‘Pre-Planorbis Beds’ (i.e. Tilmanni Ammonite Zone) of the Blue Lias Formation, although it could be latest Triassic (Rhaetian). See Lomax, Massare & Mistry (2017) for more details.

Holotype. BRLSI M3553, a partial skull, pectoral girdle and both forefins, preserved in ventral view.

Paratypes. BRLSI M3555, a skull and partial skeleton, preserved in right lateral view; BRLSI M3563, a composite partial skeleton; LEICT G454.1951/164, a partial forefin, presently missing, which might be a hindfin of a different genus (see Lomax, Massare & Mistry, 2017 for more details).

Referred specimen. BMT 1955.G35.1, an almost complete, three-dimensional skull and partial postcranial skeleton.

Emended diagnosis. As in Lomax, Massare & Mistry (2017), but with the following change: total length greater than 3.2 m but probably less than 4 m.

Occurrence. Fell Mill Farm, between Shipston-on-Stour and Honington, Warwickshire, England, grid reference NGR SP 277 415. The specimen was collected from blue-grey Liassic clay, and specifically from the Semicostatum Ammonite Zone, lower Sinemurian, Lower Jurassic.

Results

Anatomical description of the skull roof

Measurements of the skull are presented in Table 1. In lateral view, the upper jaw is shaped like a right-angle triangle, the ventral margin being nearly straight and dorsal surface of the snout being gently sloped (Fig. 1). In dorsal and ventral views, the anterior snout (formed by the premaxillae) is shaped like a finely pointed triangle (Fig. 2); the posterior portion of the skull is transversely expanded. Preserved bones of the skull roof (Figs. 1, 2 and 5) include most of the premaxillae, both maxillae, partial nasals, partial left lacrimal, partial prefrontals and postfrontals, complete left and partial right jugals, nearly complete parietals and partial supratemporals. Some of these elements (e.g. portions of nasal and postfrontals) were too fragmentary and/or poorly preserved to attach to the skull and are not part of the 3D model. The left postorbital was originally present (Fig. 1A), but we were unable to locate the element. The quadratojugals and squamosals are not preserved in BMT 1955.G35.1. The frontals are also missing with the exception of a small fragment attached to the left nasal. Unless otherwise stated, the morphology concurs with other specimens of the species (Lomax, Massare & Mistry, 2017; Lomax & Massare, 2018).

Table 1 Measurements of some skull and postcranial elements of BMT 1955.G35.1, Protoichthyosaurus prostaxalis.

Element	(cm)	
Skull length	80*	
Maxilla length	25.5R 24.2L*	
Lower jaw length	87*	
Basisphenoid length	5.82	
Basisphenoid width	9.95	
Supraoccipital height	5.04	
Supraoccipital width	6.11	
Quadrate length	9.4	
Quadrate max width	8.2	
Hyoid length	18.5R 18.2L	
Coracoid med-lat length	12.16	
Coracoid ant-post	13.66	
Scapula preserved length	12.9*	
Scapula proximal end only	7.25	
Humerus length	10.4	
Humerus distal width	8.59*	
Humerus proximal width	7.66	
Femur length	8.7	
Femur distal width	5.1	
Femur proximal width	2.5*	
Ilium length	9.38	
Humerus/femur ratio	1.2	
Notes:

‘Width’ for fin elements refers to the anteroposterior dimension, perpendicular to the long axis of the fin. L and R denote measurement of left or right elements.

Asterisk denotes an estimate because the bone is damaged or elements are missing.

Figure 5 Elements of the skull, palate, lower jaw and dentition of BMT 1955.G35.1, Protoichthyosaurus prostaxalis.

(A)–(D) Incomplete and damaged, articulated parietals in dorsal (A), ventral (B), posterior (C) and anterior (D) view. (E)–(F) Incomplete and damaged left pterygoid in posterior (E) and ventral (F) view. Note the three wing-like projections in posterior view. (G)–(I) Incomplete and damaged left quadrate in anterior (G), posterior (H) and lateral (I) view. (J) Hyoids in dorsal view. (K) and (L) Incomplete and damaged right nasal in dorsal (K) and (L) lateral view. (M) Incomplete and damaged right postfrontal in dorsal view. (N) Practically complete tooth missing the tip of the crown. Note that the root is large with prominent grooves that extend to the base of the crown and continue as longitudinal striations on the crown. Abbreviations: ac, articular condyle; (?)ce, impression of cerebellum; ch, impression of cerebral hemisphere; dpf, descending parietal flange; eed, extra-encephalic depression; ocl, occipital lamella; ol, impression of optic lobe; op, elongate openings in the posterior surface of the parietal; par, palatal ramus; ps, parietal shelf (ridge); qf, quadratojugal facet; rp, resorption pit; sc, sagittal crest; spt, supratemporal probably fused with parietals; vs, ventral surface. Scale bars represent three cm.

Premaxilla. The premaxilla makes up two-thirds of the length of the cranium and most of the snout. The majority of both premaxillae are preserved, although portions of the posterior ends are missing including the margin of the external naris (Figs. 1 and 2). The left premaxilla is more complete than the right element. In lateral view, the anterior premaxilla is dorsoventrally low but becomes progressively taller posteriorly. A longitudinal groove exposing a series of foramina (see below) along the lateral surface represents the fossa praemaxillaris (Figs. 1B, 1C and 2). The right premaxilla preserves a long, tapering subnarial process that articulates with the maxilla and extends to the middle of the maxilla (Figs. 1B and 2A); the supranarial process is not preserved on either side. Laterally, the contact between the premaxilla and maxilla is clear and consists of an extensive scarf joint in which the ventral margin of the premaxilla laterally and dorsally overlaps the anterior process of the maxilla (Figs. 1 and 2). The contact between the premaxilla and maxilla on the palate is difficult to discern, although it appears that a maxillary shelf extends medially and replaces the premaxillary shelf at the level of the 18th preserved tooth on the right side. The teeth were reset during conservation and their positions in the jaw are not original. However, their reconstructed positions act as landmarks for our description. Except at the anterior tip of the snout, the premaxillae do not meet at the ventral midline.

In dorsal view, the premaxillae would have contacted each other at a butt joint for much of their length, although they are largely separated due to deformation (Fig. 2E). Posteriorly, the nasals inserted between the premaxillae. The dorsal margin of the left premaxilla laterally and dorsally overlaps the nasal from approximately the level of the 13th premaxillary tooth to its broken posterior end. In dorsal view, a small, narrow portion of the anterior process of the nasal is exposed; the rest is overlapped by the premaxilla.

Anteriorly, the premaxilla is a laterally bowed sheet of bone in transverse cross-section; at the level of the seventh preserved tooth, it develops a medial shelf that roofs the alveolar groove. From this point until its articulation with the maxilla, the premaxilla consists of a ventral lamina that laterally overlaps the teeth, the medial shelf and a dorsal lamina, which is deeply grooved along its margin (as preserved on the right premaxilla), presumably to receive the nasal. CT scans reveal that each premaxilla encloses a branching, longitudinal canal dorsal to the tooth row (Figs. 2G–2J). This canal extends from the posterior end of the premaxillary tooth row to the third premaxillary tooth. Anteriorly, a series of short canals branch anterolaterally from the main conduit and open onto the fossa praemaxillaris, either immediately above the alveolar margin or on the dorsolateral aspect of the bone. The right premaxilla preserves five ventral and four dorsal foramina; the left premaxilla preserves four ventral and one dorsal foramina. The posterior half of each premaxilla contains two longer canals branching posteriorly from the main conduit, each of which opens onto posteriorly elongated grooves parallel to the alveolar margin of the premaxilla. The left premaxilla preserves two additional longitudinal grooves on the posterior half of its dorsolateral surface; however, these do not connect to the main canal within the premaxilla. These vascular canals within the premaxilla (as well as those within the dentary, see below) resemble canals in the facial bones of extant crocodilians and lepidosaurs, as well as those reported in theropod dinosaurs (Dal Sasso, Maganuco & Cioffi, 2009), pliosaurs (Ketchum & Benson, 2011; Foffa et al., 2014b) and plesiosaurs (Ketchum & Smith, 2010) and something similar in ichthyosaurs (Lomax & Massare, 2015). In extant taxa, these canals carry neurovascular bundles consisting of the maxillary artery and maxillary branch of the trigeminal nerve (CN V2) in the upper jaw, and the inferior alveolar artery and mandibular branch of the trigeminal nerve (CN V3) (Witmer, 1997). The complex web of ramifications reported in the upper jaw of pliosaurs cannot be visualized in this specimen; this may be due to their absence, preservation or scan resolution. Nonetheless, it is possible these canals were also associated with pressure or electro-reception as seen in some extant taxa and as postulated for dinosaurs and pliosaurs (Foffa et al., 2014b).

Maxilla. Both maxillae are preserved, although the posterior portion of the left maxilla is missing and both are damaged. In lateral view, the maxilla is a triangular bone with slender anterior and posterior processes and is dorsoventrally tallest in its centre (Figs. 1 and 2). The anterior process is longer and more delicate than the posterior process, which extends just under the orbit. Although the external naris is not preserved, it is clear the maxilla extended well beyond the anterior end of the external naris.

The alveolar groove of the maxilla is continuous with that of the premaxilla. In transverse section, the anterior maxilla has a ventral lamina that extends lateral to the tooth row, a ventrally curving medial shelf (forming the dorsal and medial walls of the alveolar groove) and a short dorsal lamina that contacts the medial surface of the premaxilla in a scarf joint. The dorsal lamina of the maxilla, which underlaps the premaxilla, is exposed slightly anterior to the middle of the left maxilla due to the damaged premaxilla. Posterior to the main body, the maxilla is triangular in transverse section with a ridge on its dorsomedial surface that appears to articulate with the short anterior process of the lacrimal, which is poorly preserved. An articulation surface on the dorsolateral surface of the posterior process of the maxilla meets the jugal in a scarf joint, separating the posterior process of the maxilla from the lacrimal.

Nasal. The anterolateral portion of the left nasal is preserved attached to the premaxilla (Figs. 1 and 2). It is best seen in ventral and posterior views, which reveals it is dorsoventrally thickened medially but becomes dorsoventrally thin laterally. The bone is laterally bowed in transverse section. The ventral margin of the nasal is laterally overlapped by the dorsal lamina of the premaxilla; the morphology of the right premaxilla suggests this may have originally been a tongue-and-groove contact. Near the posterior end of the element is a small fragment featuring a grooved medial margin; it is unclear if this is a portion of the nasal or a fragment of the frontal. CT scans reveal a few short canals penetrating the nasal from its lateral surface.

Other fragments of the nasal were found with the specimen but not mounted on the skull due to their fragile nature. Although very fragmentary, much of the right nasal is preserved although the posterior end is missing and it is impossible to determine the presence of an internasal foramen. It is a long and delicate element that is wide posteriorly, and tapers to a point anteriorly (Figs. 5K and 5L). On the medial surface is a long groove that runs almost the entire length of the nasal. The slightly flared lateral wing is damaged. Two foramina are present posteriorly, positioned next to a portion of what may be the prefrontal.

Lacrimal. The left lacrimal is poorly preserved. It appears to be triradiate with a short, but damaged anterior process and a longer posteroventral process. The dorsal process is tall and formed the posterior margin of the external naris. It was clearly excluded from the orbital margin by the anterior process of the prefrontal (Figs. 1B, 2B and 2D). The lateral surface of the dorsal process preserves external sculpting and several canals that penetrate the bone but cannot be traced. The short, tapering anterior process fits onto a shelf on the dorsomedial aspect of the maxilla. The posteroventral process, which is longer and mediolaterally wider than the anterior process, is complete and contributes to the anteroventral margin of the orbit. It meets the dorsal margin of the jugal in a curving contact. The lateral surface of the posteroventral process bears the remnant of a ridge from its posterior tip to the base of the dorsal process.

Prefrontal. Only a small portion of the anterior process of the left prefrontal is present, although original photographs of the mounted skull show that the element was once complete (Figs. 1B and 2B). The anterior process of the prefrontal medially and dorsally laps the lacrimal along a broad contact, where it is dorsoventrally tall and excludes the dorsal process of the lacrimal from the orbital margin.

Postfrontal. The anterior portions of both postfrontals are preserved but were not added to the mount. The right postfrontal is the more complete of the two elements (Fig. 5M). In dorsal view, the anterior end is mediolaterally broad and dorsoventrally thin. The postfrontal narrows posteriorly, where it is damaged. The medial surface exhibits a prominent ridge.

Jugal. The jugal is a long, slender bone forming the ventral margin of the orbit; the left is better preserved than the right (Figs. 1 and 2). Anteriorly it is oval-shaped in transverse section and tapers to a point, contacting the posteroventral margin of the lacrimal and dorsolateral aspect of the posterior process of the maxilla as previously described. Although damaged and perhaps missing a small portion, it is clear the anterior process extended to at least the level of the anterior margin of the orbit. Posteriorly, the dorsal ramus of the jugal gently curves dorsally, expands dorsoventrally and thins mediolaterally. Based on the original reconstruction (Fig. 1A), which featured a complete jugal and postorbital, the jugal contributed to about half of the posterior orbital margin.

Postorbital. An original photograph shows that the postorbital was complete, but we have been unable to locate the element (Fig. 1A). However, based on the photograph, it is clear that the postorbital is dorsoventrally short and anteroposteriorly wide, being almost rectangular in shape and making up half of the posterior orbital margin. The anterodorsal edge tapers to a narrow process.

Parietal. Both parietals are damaged and missing their anteroventral margins, the left element being better preserved (Figs. 3 and 5A–5D). In dorsal view, the parietals are hour-glass shaped and meet medially, diverging slightly anteriorly. CT scans reveal the dorsomedial margin of the anterior parietal is strongly dorsoventrally expanded in transverse section, the elements contacting each other at a tall midline butt joint; the parietal thins ventrolaterally in transverse section. The articulation of the parietals results in a well-defined sagittal crest (Figs. 5A and 5C); at its midsection, the parietal is L-shaped in transverse section with the horizontal leg forming the roof of the braincase while the ventral leg forms the lateral wall of the braincase and medial wall of the supratemporal fenestra. Lateral to the crest, the dorsal surface of the parietal is convex and curves ventrally, widening posteriorly. Posteriorly, the crest decreases in height to form an extensive shelf (parietal ridge) under which the supraoccipital articulates (Figs. 5A and 5C). Two elongate depressions, one on the posterior aspect of each parietal, may represent attachment sites for epaxial neck muscles (Fig. 5C).

In ventral view, the surface of the parietal is concave and bears impressions of structures that surrounded the brain (Figs. 5B and 5D). In the anterior region, impressions of the cerebral hemisphere and extra-encephalic depression are present (as in McGowan, 1973). McGowan (1973, fig. 48) showed that the cerebral hemisphere was present in both the parietal and frontal in a specimen of Ichthyosaurus. In BMT 1955.G35.1, there is no indication of the frontal at this position, suggesting the cerebral hemispheres were likely limited to the parietal. The descending parietal flange is present in both parietals, although the left is more complete (Figs. 5B and 5D). The anterior process is thick, short and protrudes forwards, creating a ledge. Towards the centre of the parietal is the large, ovoid impression of the optic lobe, the most prominent of the cerebral structures, situated posterior to the parietal flange (Fig. 5B). The epiterygoid process is not preserved. Posteriorly, the parietal flares laterally to form the paraoccipital process; in posterior view, this process is shaped like a bowtie and ventrally deflected. In ventral view, there may be an impression of the cerebellum, although this is difficult to confirm because this portion is damaged.

Supratemporal. Portions of both supratemporals are preserved. The majority is exposed at the posterior margin of the skull, attached to the parietal (Figs. 3C and 5C). It is difficult to identify the parietal–supratemporal suture in the original specimen. In CT scans, the contact between the left parietal and supratemporal is visible as a very tight, sinuous butt joint; this contact cannot be discerned on the right and the two bones may have fused. In posterior view, the preserved supratemporal is large and triradiate; it is narrow medially and increases in width distolaterally, with a posteroventral process. In this view, it is roughened with numerous striae, probably for muscle attachment (Kirton, 1983) (Fig. 5C). There are also some foramina present, similar to those reported in this region of the supratemporal in ichthyosaurs such as the Cretaceous Leninia stellans (Fischer et al., 2014).

Anatomical description of the palate

The left pterygoid, including a fragment representing the quadrate wing, and quadrate are preserved (Fig. 3).

Pterygoid. The left pterygoid can be positively identified, although it is damaged. It is an anteroposteriorly elongate element with a robust and mediolaterally wide posterior end and narrow anterior end (palatal ramus) (Figs. 3, 5E and 5F). The palatal ramus is dorsoventrally flattened and makes up over half the length of the pterygoid; it is narrowest at its midlength and expands distally. Posteriorly, the pterygoid expands transversely and dorsoventrally to form the quadrate ramus; its dorsal surface rises in a ridge that would have been continuous with the quadrate wing (see below). Although damaged and incomplete, the overall morphology of this element, particularly how the shape changes from the posterior end to the narrow midshaft which then broadens anteriorly, is reminiscent to the pterygoid of Sveltonectes (Fischer et al., 2011, fig. 2G). This differs from Ichthyosaurus, which has a very narrow shaft posteriorly (McGowan, 1973, fig. 20), Platypterygius longmani Wade, 1990 which has a mediolaterally wider shaft (Kear, 2005, fig. 8C–8E) and Ophthalmosaurus icenicus Seeley, 1874, in which the pterygoid has a distinctly different shape posteriorly (Moon & Kirton, 2016, plate 6, figs. 1, 2).

In dorsal view, the posterior end has three wing-like projections. The medial projection, which is damaged and was originally more extensive, is the largest and most robust, whereas the lateral projection is slender and dorsoventrally flattened (Fig. 5E). The ventral surface is better preserved, although the edge of the interpterygoid vacuity is damaged (Fig. 5F). Regardless, the posterior end of the pterygoid is larger, wider and narrows more gradually than that of Ichthyosaurus (McGowan, 1973, fig. 20B). The dorsal (quadrate) wing of the posterior ramus of the left pterygoid is almost certainly represented by a large but thin fragment of bone, the shape of which was obscured by a large amount of wood and plaster in the original reconstruction but is revealed in CT scans.

Quadrate. Only the left quadrate is preserved, which is a large and robust element (Figs. 3 and 5G–5I). In anterior and posterior views the quadrate is C-shaped, owing to strong curvature of the shaft (Figs. 5G and 5H); it is more of an L-shape in Ichthyosaurus (McGowan, 1973, fig. 9). The articular condyle is massive and greatly expanded mediolaterally, whereas the dorsal end is mediolaterally thin. A well-defined ridge is present above the condyle and displays a long groove identified as the quadratojugal facet. A groove is present on the ventral surface of the condyle, which divides the jaw joint surface into two distinct faces, is common among ichthyosaurs.

Anatomical description of the braincase

Preserved material includes the supraoccipital, left opisthotic, left stapes and parabasisphenoid (Fig. 3). The anterior portion of the parasphenoid as well as the basioccipital, prootics and exoccipitals are missing.

Supraoccipital. The median supraoccipital is triangular with its apex anterodorsally directed (Figs. 6A–6C). CT scans revealed that the right margin of the supraoccipital had been reconstructed in plaster, obscuring the true shape of this element. In anterior and posterior views, the element is convex and arch-like, and is wider than it is tall, which is similar in Ichthyosaurus (McGowan, 1973, fig. 4). Of particular note, in this view, the dorsal portion of the opening for the foramen magnum is much more reduced than in either Platypterygius longmani (Kear, 2005, fig. 10D–10E) or Ophthalmosaurus icenicus (Moon & Kirton, 2016, plate 9, fig. 1–5). A median ridge is present on the posterior surface, which is sharpest anterodorsally and flattens as it approaches the foramen magnum (Figs. 6B and 6C). This ridge would have contacted the parietal, as shown in the 3D model (Figs. 3C and 3F) and separates two flat, posterolaterally-directed faces, each of which is pierced by a canal that opens onto its internal surface (Figs. 3B and 3G). These openings probably represent the foramen endolymphaticum (Andrew, 1910), which served for the passage of the endolymphatic ducts (McGowan, 1973; Maisch, 2002; Marek et al., 2015) or veins (Kirton, 1983; Moon & Kirton, 2016). The complete left half preserves two articulation facets along its ventral lateral margin—a larger, posteroventrally-directed facet that is deep and triangular-shaped (apex pointing forward) and a smaller, oval-shaped facet, that is, posterolaterally-directed.

Figure 6 Braincase elements of BMT 1955.G35.1, Protoichthyosaurus prostaxalis.

(A)–(C) Incomplete supraoccipital in posterior (A), dorsal (B) and ventral (C) view. (D) and (E) Parabasisphenoid with complete basisphenoid and broken parasphenoid in anterior (D) and ventral (E) view. (F) and (G) Left opisthotic in anteromedial (F) and ventrolateral (G) view. Note the ‘V-shaped’ membranous impression in (F). (H) Incomplete left stapes in posterior view. Abbreviations: bf, facet for basipterygoid facet; bof, basioccipital facet; bp, basipterygoid process; cf, carotid foramen; ds, dorsum sellae; ef, exoccipital facet; hsc, horizontal semicircular canal; (?)ma, muscle attachment; mh, medial head; mr, median ridge; p, base of parasphenoid; pp, paroccipital process; pvsc, posterior vertical semicircular canal; rfm, roof of foramen magnum; sac, sacculus; sf, stapedial facet; st, sella turcica; t, paired trabeculae; tg, trenchant groove; (?)ut, utriculus. Scale bars represent three cm.

In dorsal view, there is a well-defined ridge that is separated by a long, trenchant groove (Fig. 6B). For Ichthyosaurus, McGowan (1973, p. 15) described the dorsal edge as having two shallow grooves. The groove marks the boundary between the ossified and cartilaginous portions of the neurocranium (McGowan, 1973). In ventral view, the element is arched with a smooth section for the roof of the foramen magnum (Fig. 6C). The exoccipital facet is roughly square.

Parabasisphenoid. The thin parasphenoid is broken with a small portion preserved fused to the basisphenoid (Fig. 6D). The basisphenoid is complete and is a large, robust element both mediolaterally wide and dorsoventrally tall (Figs. 6D and 6E). There are deep grooves between the posterior corners of the bases of the basipterygoid processes and the main body for the palatal ramus of the facial nerve (Kirton, 1983). In dorsal view, the midline of the anterior end is convex and, along with the protruding anterior ends of the basipterygoid processes, gives the anterior margin of the basisphenoid a ‘three-pronged’ appearance, resembling a specimen of Ichthyosaurus referred to as the ‘Evans Nodule’ by McGowan (1973, plate 1a). The basipterygoid processes are both complete, robust and oblong in ventral view (Fig. 6E). Their surfaces appear slightly roughened, probably due to a cartilaginous covering for contact with the pterygoid. The distal articular facet of the basipterygoid process is defined by a depression with a rim. The anterior tip of the basipterygoid process is tapered, whereas the posterior margin is thickened and rounded.

The anterodorsal aspect of the basisphenoid features a pair of robust protuberances separated by a slight midline depression—the sella turcica—that housed the pituitary gland (Fig. 6D). Below this is the median opening for the carotid artery, which courses posteroventrally through the bone and exits on its ventral surface as a rounded opening bounded proximally by an arch-like ridge (Figs. 6D and 6E). Ventral to this opening and dorsal to the parasphenoid is a kidney-shaped articulation facet, interpreted as the impressions of paired trabeculae (as in McGowan, 1973, fig. 1) (Fig. 6D). Immediately dorsal and posterior to the sella turcica, is a large, bulbous region that has the ossified dorsum sellae (dorsal crest). The posterior surface is a wide, rounded rectangle, indented for reception of the basioccipital.

Opisthotic. Only the left opisthotic could be identified (Figs. 6F and 6G). It is a robust and stout element that is roughly pentagonal in posterior view. Its ventrolateral margin is long and sharp. Ventrally the opisthotic tapers to a point that bears a small facet, which articulates with the stapes. The stapedial facet is large, but the lateral ‘foot’ (after Fischer et al., 2012) has minor exposure. The ventromedial margin is concave and bears a long, low groove that marks the basioccipital facet (Fig. 6G). The dorsolateral margin forms the prominent paroccipital process, the posterior surface of which bears a long, prominent ridge that ascends vertically from the ventral tip of the element, then turns medially. A deep groove, for either the glossopharyngeal or branch of the facial nerve (Kirton, 1983; Marek et al., 2015), separates this ridge from a pronounced protuberance on the dorsal margin of the opisthotic. The dorsomedial margin is expanded into a rugose, subtriangular depression (apex pointing posterodorsally) surrounded by a raised lip and several small protuberances. Although poorly preserved, the membranous impressions of the posterior vertical semicircular canal, sacculus, the horizontal semicircular canal and possibly utriculus are represented by a somewhat ‘V-shaped’ impression, best observed in anteromedial aspect (Fig. 6F). The impression of the horizontal semicircular canal is damaged at the tip and the impression of the sacculus is wide and round. There are several grooves positioned adjacent to the impressions, which McGowan (1973, fig. 5) referred to as grooves in the margin circumscribing the membranous impression. CT reveals a great deal of trabecular bone within the opisthotic.

Stapes. Both stapes are preserved, with the left being more complete. The stapes is mediolaterally elongate with a bulbous occipital head and a tapered distal end (Fig. 6H). The proximodorsal region of the medial head bears a groove that marks the course of the stapedial artery. In anterior view, the medial head is laterally inclined and there is a shallow groove, which is probably the opisthotic facet. The posterior surface of the stapes bears a series of oblique ridges and grooves. This may have been an area for muscle attachment (McGowan, 1973, fig. 7A) (Fig. 6H). There are several small canals within the stapes; however, these are very difficult to trace.

Anatomical description of the lower jaw

Nearly complete left and right dentaries are present, as are both incomplete splenials, the nearly complete left surangular and the complete left articular and angular (Fig. 4).

Dentary. The dentary makes up over three-quarters the length of the lower jaw. It is elongate, tapering at its anterior and posterior ends (Figs. 1 and 4). The ventral margin is convex while the dorsal margin is concave, and the entire element curves dorsally at its anterior end; the latter is likely the result of taphonomic distortion. As with the upper jaw, the lower teeth have been reset in a continuous groove, which we use as landmarks for our description. In transverse section, the anterior dentary is roughly oval-shaped with a convex lateral surface; a medial shelf forms the floor of the alveolar groove and a dorsal lamina laterally overlaps the dentary teeth. The medial shelf is separated from a longitudinal ridge that parallels the ventral margin of the bone by a shallow groove (lateral wall of the Meckelian canal); this ridge and groove dominate the internal face of the anterior half of the dentary. At the level of the 15th dentary tooth, the medial shelf disappears and the dentary becomes a laterally bowed sheet of bone with a thickened dorsal margin in transverse section.

The anterior tip of the right dentary is damaged and, as a result, the dentaries do not contact each other anteriorly to form the mandibular symphysis (Figs. 1 and 4). As preserved, the dentary and splenial do not contact each other along their entire length but this is due to distortion. The anterior tip of the angular is level with the 17th preserved tooth on the right side; the angular laterally overlaps the ventral margin of the dentary in a very tight scarf joint. In contrast, the suture between the dentary and surangular, which reaches the level of the 22nd preserved dentary tooth, is a loose, horizontal butt joint except at its posterior end where the posterior tip of the dentary laterally overlaps the surangular.

As with the premaxilla, CT scans reveal that each dentary encloses an elongate, branching canal ventral and lateral to the tooth row that extends from the anterior tip of the bone to the 14th (right) and ninth (left) preserved dentary teeth, at which point the canal opens onto the internal surface (Meckelian canal) of the lower jaw ventral to the medial shelf of the dentary (Figs. 4C and 4G–4J). Anteriorly, four small canals branch laterally from the main conduit and open onto short, posteriorly elongated grooves on the lateral face of the dentary. A posterior (fifth) canal opens into a very long groove ventral and parallel to the tooth row that extends over a quarter the length of the dentary.

Splenial. The splenial is composed of a vertical sheet of bone that is medially concave, a slightly thickened dorsal margin that is turned medially, and a thickened, laterally deflected ventral margin. Thus, the element has a mild S-shape and is mediolaterally thin in transverse section anteriorly, becoming more robust with increasingly pronounced curvature posteriorly. The splenial forms the medial wall and part of the floor of the Meckelian canal for the posterior half of the lower jaw. Its contacts with other elements cannot be reliably interpreted as the bones were not in articulation; however, from their preserved ventral margins, it appears the splenial and angular met in a butt joint.

Angular. The angular extends over half the length of the lower jaw (Figs. 1 and 4B). The anterior half of the angular is a long, straight rod while the posterior half is both dorsoventrally and mediolaterally expanded, curving dorsally and medially towards the jaw joint. In transverse section, the anterior half of the angular is diamond-shaped with a dorsomedial surface that contacts the ventral margin of the dentary in a tight scarf joint and a dorsolateral surface that meets the ventral margin of the surangular in a loose butt joint. The ventromedial surface of the anterior angular bears a shallow, longitudinal groove bounded dorsally and ventrally by low ridges that presumably articulated with the splenial. Posteriorly, the angular develops a robust tab or lamina that extends from its dorsomedial surface and medially laps the surangular. However, immediately ventral to the jaw joint, this lamina disappears and is replaced by taller, mediolaterally thin dorsolateral lamina that extensively overlaps the lateral aspect of the posterior surangular. Thus, the contact between the angular and surangular is morphologically simple and loose anteriorly but tighter and more complex posteriorly. In lateral view, the anterior end of the surangular is broken and it appears the angular extends further anteriorly than the surangular (Fig. 4B). This is similar to specimen SOMAG 12, a referred specimen of P. prostaxalis (Lomax, Massare & Mistry, 2017).

Surangular. The surangular is a long, curved element forming the lateral aspect of the posterior third of the lower jaw (Figs. 1 and 4B). The anterior half of the surangular is poorly preserved as it is mediolaterally thin and is loosely joined to the dentary (dorsally) and angular (ventrally) via rounded butt joints. Posterior to the dentary, the dorsal margin of the surangular thickens dramatically to form the peaked coronoid process. A longitudinal lateral ridge, dorsally bounding the fossa surangularis, continues to the end of the surangular and separates the thickened dorsal margin from the thinner ventral lamina that articulates with the angular. The element expands dorsally and medially at its rounded posterior end to laterally cup the articular.

In medial view, the posterior surangular bears a ridge parallel to its ventral margin that articulates with the angular and forms the floor of the adductor fossa. There is another, more robust ridge on the medial surface originating at the coronoid process and widening posteriorly to contact the anterior surface of the articular. The medial face of the surangular between the two ridges is concave and forms the Meckelian groove and lateral wall of the adductor fossa. There is a large foramen clearly visible on the medial aspect ventral to the coronoid process; this foramen passes laterally through the surangular and exits ventral to the ridge on the lateral surface (Figs. 4D and 4K).

Articular. The preserved left articular has a triangular profile in dorsal and ventral views, with the apex posteriorly and medially directed, and a subcircular profile in medial and lateral views. The posterior margin is sharp while the anterior aspect is flat and broad where it contacts the quadrate to form the jaw joint. The medial aspect of the bone is smooth while the lateral aspect is pitted and porous. CT scans reveal several small, short canals that penetrate into the bone from its lateral surface.

Hyoid. Both hyoids are preserved and are large and complete, although some damage is apparent. The hyoid is a curved, rod-like bone (Fig. 5J). In dorsal view, the element is slightly bowed posterolaterally and the centre of the element is slightly mediolaterally narrower than either end. The anterior end is slightly flattened, rounded and pitted for reception of cartilage. In anterior view, the probable left hyoid is oval-shaped, with a defined rim.

Dentition. The teeth were implanted in an aulacodont fashion in continuous alveolar grooves as is typical in euichthyosaurs. As previously mentioned, the teeth were not preserved in situ and were added to the grooves during reconstruction of the skull both in 1955 and 2015; thus, they are not in their original positions. Furthermore, the dental groove is too poorly preserved to determine the exact number of teeth that would have originally been present. There are additional fragmentary and complete teeth associated with the specimen.

The teeth are lingually curved, large cones with short, robust crowns with fine striations and smooth apices (Figs. 1B, 1C and 5N). In complete teeth, the crown is much narrower than the root. The roots are large with prominent longitudinal grooves that extend to the base of the crown and continue as longitudinal striations on the crown (Fig. 5N). This morphology is found in all specimens of Protoichthyosaurus that have well-preserved teeth (Lomax, Massare & Mistry, 2017; Lomax & Massare, 2018). Tooth morphology for each tooth is similar, with crowns ranging from 0.87 to 1.75 cm in height. As no teeth were preserved in situ, it is impossible to differentiate between the premaxillary, maxillary and dentary teeth. A resorption pit is present on the lingual surface in many teeth (Fig. 5N). CT scans reveal hollow pulp cavities within the teeth that open at the tooth bases and extend nearly the entire height of the tooth.

Anatomy of the postcranial skeleton

Portions of the vertebral column, ribs, gastralia, forefin, pectoral girdle, pelvic girdle and the hindfin are preserved (Fig. 7). The forefin and hindfin phalangeal elements are entirely free of matrix and are not in their original context, so it is impossible to say whether elements are from the left or right fin.

Figure 7 Elements of the postcranial skeleton of BMT 1955.G35.1, Protoichthyosaurus prostaxalis.

(A) and (B) Probable ‘unfused’ (see text for details) axis vertebra in anterior (A) and ventral (B) view. Note the unusual, almost rugose anterior surface that is rarely seen in ichthyosaurs. The dark, circular element to the right is a poorly preserved bivalve mollusc. (C) Left coracoid in dorsal view. (D) Incomplete left scapula in lateral view. (E) and (F) Left humerus in dorsal (E) and ventral (F) view. Note that the dorsal process (trochanter dorsalis) is damaged, as is the facet for the ulna. (G) Complete ilium in either lateral or medial view. Note that the posterior end (to the right) is bulbous, relative to the shaft. (H) and (I) damaged (?)right femur in dorsal (H) and ventral (view). Abbreviations: af, anterior facet; aif, facet for the axial intercentrum; an, anterior notch; bpe, broken posterior end; bpe, bulbous posterior end; ccf, facet for the cervical centrum; dp, dorsal process; dpc, deltopectoral crest; ff, fibular facet; gf, glenoid facet; if, intercoracoid facet; pm?, predation marks; pn, posterior notch; rf, radial facet; sf, scapular facet; tf, tibial facet; uf, ulnar facet; vp, ventral process. Scale bars represent three cm.

Axial skeleton. A total of 37 vertebral centra are present, all of which are disarticulated. Most are poorly preserved but their positions in the column can be identified from their morphology. One centrum is unusual in possessing the following features: triangular in anterior and posterior views; being marginally anteroposteriorly longer than the preserved cervicals; diapophyses and parapopthyses being high and positioned at the anterior end of the centrum in lateral view; two separate semicircular facets for articulation with intercentra in ventral view (Figs. 7A and 7B). This morphology is indicative of an atlas-axis complex, but the centrum displays no fusion. This is unusual given that, with the possible exception of immature individuals and some early Triassic taxa, the atlas-axis is always fused in ichthyosaurs (McGowan & Motani, 2003; VanBuren & Evans, 2017). The presence of two facets on the ventral surface might suggest that this element is the atlas, with the diagonally-oriented anterior facet being for the atlantal intercentrum and the posterior facet for the axial intercentrum (Fig. 7B). Alternatively, and more likely, this is the axis, with the anterior facet being for the axial intercentrum and the posterior facet being for the intercentrum of the third cervical vertebra (McGowan & Motani, 2003, fig. 5C). Interestingly, the anterior surface of the axis centrum is not well-defined, nor smooth and lacks the convexity typical of ichthyosaur centra (Fig. 7A). This might be pathological or it could be the surface that was fused with the atlas vertebra that is not usually preserved (or exposed). A second centrum features similar morphology but is slightly anteroposteriorly shorter and has only one small, anterior facet on the ventral surface, which articulates with the aforementioned vertebra. It is likely that this is the centrum of the third cervical vertebra. The remaining vertebral centra include 19 dorsals, including elements from the anterior, middle and posterior portions of the series as identified by their shape and position of the diapophyses and parapopthyses, and 16 caudal vertebra, again including elements from the anterior, middle and posterior portions of the series as identified by their shape and the presence of a single rib facet.

One isolated and damaged neural spine, which is mediolaterally thin at its distal end, is preserved.

Numerous incomplete ribs and rib fragments are preserved. The cross-sectional geometry of the ribs varies, with some being rounded whereas others have a dumbbell-shaped cross section. A possible gastralia fragment is present, which is roughened at its anterior end where it presumably met its counterpart at the midline.

Pectoral girdle. The left coracoid is practically complete (Fig. 7C). It is a robust element that is slightly anteroposteriorly longer than mediolaterally wide (Table 1). It has prominent and well-developed anterior and posterior notches. The anterior notch is wider than the posterior notch, resulting in the posterior end of the coracoid being mediolaterally wider than the anterior end. A prominent rim outlines the glenoid and scapular facets, the former being noticeably longer than the latter. In medial view, the intercoracoid facet is dorsoventrally thickened and bulbous at the anterior end but narrows posteriorly.

Only the left scapula is preserved and is missing its posterior end (Fig. 7D). The anterodorsal end is marked by a right angle, which extends to the ventral edge. This proximal end is twice as tall dorsoventrally as the mid shaft and is widely flared but without a prominent acromion process.

Forefin. As mentioned previously, none of the phalangeal elements were found in articulation. It is impossible to determine whether the elements are from the left or right fin or determine the morphology of the forefin in this specimen. The radius and ulna are missing and the preserved elements are polygonal. Of note, the forefin was reconstructed for display in 1955 and 2015 with the morphology typical of Ichthyosaurus (Motani, 1999). This was prior to the resurrection of Protoichthyosaurus (Lomax, Massare & Mistry, 2017).

A single, nearly complete left humerus is robust, elongate and slightly wider distally than proximally without a prominent constriction in the mid shaft (Figs. 7E and 7F). It is the largest humerus of Protoichthyosaurus described thus far (Table 1). The proximal end is large, bulky and the surface is rugose and roughened. In ventral view, the deltopectoral crest is offset anteriorly and is large but does not extend far down the shaft. The base of the anterior end is slightly flared due to the presence of an anterior facet. The dorsal process is broken but appeared centrally located. There are several possible predation marks preserved on the ventral surface of the humerus (Fig. 7F). The facets for the radius and ulna are also damaged.

Pelvic girdle. A single ilium is well-preserved (Fig. 7G). It is a relatively thick and elongate element that is J-shaped in lateral and medial views, resembling the ilium of I. somersetensis in being more oblong than rib-like (Lomax & Massare, 2017). The presumed posterior end is slightly bulbous, relative to the shaft, somewhat similar to the ilium of P. applebyi (Lomax, Massare & Mistry, 2017, UNM.G.2017.1). The presumed anterior end is highly rugose. A possible ischium might also be preserved, but it is heavily damaged.

Hindfin. Like the forefin, some phalanges of the hindfin are preserved, which are largely polygonal, but none were found in articulation and all have lost their original context. Regardless, the single, incomplete femur provides information (Figs. 7H and 7I). As the proximal end is poorly preserved, it is difficult to identify the element as being from the left or right, but it is most likely a right femur, based on the following comments. It has a very slender shaft, narrow proximal end and a flared distal end. Both the dorsal and ventral processes are damaged and worn, but the supposed dorsal process seems to be a prominent, narrow ridge and the supposed ventral process is large. There is a slight flare at the anterior end, but the posterior end is only slightly expanded, and is almost a right angle. The tibial facet is larger than the fibular facet.

Historically modelled regions of the skull of BMT 1955.G35.1

Computed tomography-scanning the skull of BMT 1955.G35.1 aided substantially in our anatomical description. Additionally, modelled areas of the skull can be clearly differentiated from fossil bone in scans by the differing densities of these materials (Fig. 8). Fossil bone is the densest material (appearing as bright areas within CT scans) followed by regions of the braincase that were 3D printed in gypsum (see Material and Methods). Areas of the skull modelled during its initial reassembly post-May 1955 are the least dense, as they are either composed of wood or a traditional mix of alvar, jute and kaolin (known as AJK dough). Some modelled areas—such as the posterior third of the right lower jaw, central portion of the right jugal and ‘symphysis’ between left and right dentaries—are immediately apparent. Other areas, including the right lacrimal and prefrontal, and various patches in the lower jaws, are less obvious. The skillfully modelled right margin of the supraoccipital is only evident in CT scans, as are portions of the braincase that were 3D printed and added to the newly reassembled skull. Thus, our work demonstrates the utility of applying CT scanning to older, potentially modified museum specimens to better understand both anatomy and specimen history.

Figure 8 Surface models (generated from CT scan data) of the skull of BMT 1955.G35.1, Protoichthyosaurus prostaxalis, highlighting differences between the original skull and reconstruction.

Fossil bone (grey), regions reconstructed during original reassembly in the 1950s (yellow), and regions reconstructed in the course of the current work (blue). Right (A) and left (B) lateral, and dorsal (C) and ventral (D) views of the upper and lower jaws.

3D digital reconstruction of the skull of BMT 1955.G35.1

Limits to the data set used in the 3D digital reconstruction of the skull must be noted. Numerous bones are absent, fragmentary or were too delicate to scan, and some aspects of the 3D reconstruction are uncertain. For example, the width of the reconstructed skull is constrained by the articulation of the premaxillae (anteriorly) and contacts between the basisphenoid, pterygoids and quadrate (posteriorly). Bones of the skull roof and palate that determine width in the middle part of the skull are missing. Furthermore, the placement of the preserved bones of the posterior skull roof is an estimate based on (1) the predicted height of the missing exoccipitals relative to other braincase elements and (2) the assumption of a smooth slope between the nasals and parietals, as observed in other large ichthyosaurs, including examples of the genus Protoichthyosaurus (Lomax, Massare & Mistry, 2017; Lomax & Massare, 2018). We did not attempt to retrodeform elements that experienced plastic deformation, specifically the lower jaws. The exaggerated dorsal and lateral curvature of these elements prevents complete closure of the upper and lower jaws in our model. Similarly, the premaxilla and nasals could not be completely rearticulated due to their deformed nature. Thus, this 3D digital reconstruction is our current best hypothesis of the original skull shape of BMT 1955.G35.1 based on preservation and personal interpretation. With these limitations in mind, the digital reconstruction nonetheless yields useful new information on overall skull shape in this taxon (Fig. 9; Appendix S3). This skull shape is typical of P. prostaxalis in lateral view (Fig. 9A), in having a low skull that is slightly inclined from the nasals to the posterior end of the skull and in possessing a relatively long and slender rostrum especially when compared with Lomax, Massare & Mistry (2017, figs. 2C, 4A–B) and Lomax & Massare (2018, figs. 2–3).

Figure 9 Surface models (generated from CT scan data) of the skull of BMT 1955.G35.1, Protoichthyosaurus prostaxalis.

After the removal of minor damage and duplication/mirroring of asymmetrically preserved elements, and digital articulation of individual bones to produce a more accurate digital 3D reconstruction. Displacement of the lower jaw and premaxillae and nasals are the result of deformation (see text). Left lateral (A) dorsal (B) ventral (C) anterior (D) and posterior (E) views of the upper and lower jaws. Individual bones labelled using the same colours as Figs. 2–4.

Due to the limitations of the fragile nature of the specimen some of the bones could not be articulated in life position and there are differences between the digital model and physical specimen (Figs. 1 and 2; Appendix S1). Of note, the rear of the skull is mediolaterally wider and dorsoventrally shorter in the digital reconstruction than in the physical specimen. This is due to placement of the basisphenoid dorsal and anterior to its true articulation with the pterygoids in the physical specimen as well as midline contact between the pterygoids; the pterygoids are separated by the basisphenoid in ichthyosaurs (McGowan, 1973; Kirton, 1983; Kear, 2005). The stapes is dorsally displaced in the physical reassembly; in other ichthyosaurs, the stapes contacts the quadrate dorsal to its expanded base (Andrew, 1910; Kirton, 1983; McGowan & Motani, 2003). Lastly, the jugal extends posterior to the quadrate in the physical specimen leaving no space for the posterior facial bones and resulting in the upper jaw being anteroposteriorly shorter than the lower jaw. Shifting premaxilla and contacting bones so that the anterior tips of the premaxillae and dentaries are level results in a gap between the jugal and quadrate large enough to accommodate the missing postorbital and quadratojugal. These differences highlight another advantage of applying 3D imaging and visualization methods to large specimens. Large fossil bones are fragile and heavy, and there are practical limitations to how they can be physically manipulated and mounted when reassembling a skull or skeleton; digital manipulation of fossil bones reduces risk to the specimen and errors can be easily corrected.

Discussion

BMT 1955.G35.1 has never formally been described. The original museum record shows that it was initially identified as I. communis, a species to which many ichthyosaur specimens were historically referred as it is among the most common ichthyosaurs in the UK (but see Massare & Lomax, 2017). In notes held at the Warwickshire Geological Records Service (NRL, personal communication, J. Radley, 2015), a report by Dr Brian Seddon, stated: ‘It is believed that this animal is a new species lying somewhere between communis (I. communis) and breviceps (I. breviceps)’. A 1957 letter from Seddon states that it was ichthyosaur expert Robert Appleby who expressed the opinion that the specimen possibly represented a new species and requested photos be taken. More recently, Larkin et al. (2016) tentatively identified the specimen as Ichthyosaurus, based on available information at the time. Since then, a revised diagnosis of Ichthyosaurus has been published (Massare & Lomax, 2017), along with a redescription of Protoichthyosaurus (Lomax, Massare & Mistry, 2017), a genus first described by Appleby (1979), which was later synonymized with Ichthyosaurus (Maisch & Hungerbühler, 1997).

Lomax, Massare & Mistry (2017) provided an emended diagnosis of Protoichthyosaurus, which included several autapomorphies of the forefin. Furthermore, Lomax & Massare (2018) provided additional information on the genus and species, including revisions to the diagnosis, and showed that the genus can also be distinguished from Ichthyosaurus by a combination of skull characters. They further noted that characters used to distinguish individual species of Protoichthyosaurus from individual species of Ichthyosaurus are more easily evaluated. The forefin of BMT 1955.G35.1 is entirely reconstructed and we have been unable to locate photographs or illustrations of how the freshly excavated forefin appeared. Thus, the forefin cannot be used to identify the specimen and so identification is based upon specific skull characters.

BMT 1955.G35.1 does possess features shared by both Ichthyosaurus and Protoichthyosaurus, including: a coracoid with both prominent anterior and posterior notches; scapula with a narrow shaft that is expanded at the anterior end, but without a prominent acromion process; a humerus with nearly equal width distally and proximally, with only a slight constriction in the shaft; and femur longer than wide, with distal end wider than proximal end. However, BMT 1955.G35.1 can be assigned to Protoichthyosaurus on the basis of several characters. Some of these characters are also found in some species of Ichthyosaurus but not in the same combination (Lomax & Massare, 2018). They include: the prefrontal anterior process separates the lacrimal dorsal process from the orbit margin; strongly asymmetric maxilla with long, slender anterior process; teeth that have large roots with deep, prominent grooves that extend to the base of the crown and are continuous with the ornamentation of the crown itself; and a long, slender rostrum. In addition, the slightly diverging anterior end of the parietals in BMT 1955.G35.1, which leaves an opening at the anterior end, is indicative of the posterior opening for the pineal foramen between the parietals and frontals. Because the frontals are not preserved, it is not possible to confirm if this is correct, but it seems plausible as this is the position of the pineal in Protoichthyosaurus (Lomax & Massare, 2018). In Ichthyosaurus the pineal is between the frontals and parietals (Massare & Lomax, 2017).

Only two species of Protoichthyosaurus are considered valid, Protoichthyosaurus prostaxalis and P. applebyi, which differ in skull and humeral morphologies (Lomax, Massare & Mistry, 2017). A third questionable species, P. fortimanus, known only from an isolated forefin missing the humerus, displays only characters of the genus (see Discussion in Lomax & Massare, 2018). The left humerus of BMT 1955.G35.1 is damaged on its dorsal surface. This restricts its usefulness in identification because the two species can be differentiated by the dorsal process, which is missing in this specimen. Nevertheless, the humerus of BMT 1955.G35.1 is robust, more similar to P. prostaxalis than P. applebyi, but this may be due to the large size of BMT 1955.G35.1 (see Lomax, Massare & Mistry, 2017, fig. 5). However, considering the size, Lomax & Massare (2018) recently described the second known specimen of P. applebyi, an isolated skull (NHMUK R1164), which is comparable in size with some smaller specimens of P. prostaxalis. They identified NHMUK R1164 as probably an adult and showed that the differences among the two species are not ontogenetic. BMT 1955.G35.1 is more than twice the size of NHMUK R1164 and is probably an adult P. prostaxalis. Unfortunately, BMT 1955.G35.1 is missing some features of the skull that distinguish the two species. However, the maxilla of BMT 1955.G35.1 is large, triangular, dorsoventrally high, and possesses a long and narrow anterior process that is longer than the posterior process. Whereas in P. applebyi, the maxilla is dorsoventrally low. Furthermore, although the jugal is incomplete and the postorbital is missing, they were complete and part of the original mount (Fig. 1A). The morphology of the postorbital, although based on the interpretation of a photograph, in being dorsoventrally short but anteroposteriorly wide almost rectangular, and making up half of the posterior orbit margin are characters found in P. prostaxalis (Lomax, Massare & Mistry, 2017; Lomax & Massare, 2018). In P. applebyi, the postorbital is dorsoventrally long, anteroposteriorly narrow, and makes up much more than half of the orbit posterior margin (Lomax & Massare, 2018). Consequently, even accounting for a misinterpretation of the postorbital morphology, its shape does not match what is found in P. applebyi. Thus, based on the morphology and extent of the maxilla and postorbital, we assign the studied specimen to P. prostaxalis. The difference in size between the studied specimen and the presumed adult specimen of P. applebyi (NHMUK R1164) is another indicator that the studied specimen belongs to P. prostaxalis.

It should also be noted that the maxilla of BMT 1955.G35.1, although dorsoventrally high, does not appear as tall as in some specimens of P. prostaxalis (e.g. BRLSI 3555, BU 5323), but this is due to damage to the dorsal lamina of the maxilla on both sides. It may also appear shorter due to the length of the studied skull, which is almost twice that of the largest reported specimen of P. prostaxalis (Lomax, Massare & Mistry, 2017; Lomax & Massare, 2018), with an estimated total skull length of at least 80 cm and estimated mandible length of 87 cm. This is also much larger than the sister taxon Ichthyosaurus, with maximum skull and mandible lengths of 57.5 and 67 cm, respectively (Lomax & Sachs, 2017). Considering that the skull length is approximately 20–25% of the total body length, based on a paratype specimen of P. prostaxalis (BRLSI M3555), we estimate BMT 1955.G35.1 would have been between 3.2 and 4 m in length. This is the largest example of the genus known, the previous total length estimate being 2.5 m (Lomax, Massare & Mistry, 2017). The largest unequivocal example of Ichthyosaurus has a maximum total body length estimate of 3.3 m (Lomax & Sachs, 2017), thus the maximum length estimate of the specimen described herein is also larger than all known examples of Ichthyosaurus.

Conclusions

In this study, we describe a large, partial ichthyosaur skeleton from the Early Jurassic of Warwickshire, England. In addition to examining the specimen, we carried out CT scanning of individual skull bones as well as the entire, reassembled skull, one of the first times the skull of a large marine reptile has been successfully CT-scanned, visualized and reconstructed in 3D (see McGowan, 1989; Foffa et al., 2014a). CT scanning contributed greatly to our anatomical description by revealing features not visible on original fossil material such as: branching, longitudinal vascular canals within the premaxilla and dentary; short canals penetrating the nasal, lacrimal, stapes and articular; trabecular bone within the opisthotic; canals in the basisphenoid and supraoccipital; the presence of the quadrate process of the pterygoid; and the sutural morphology. We also demonstrate the utility of applying medical imaging techniques to historic specimens to differentiate between original fossil material and reconstructed regions, as well as the advantage of using digital visualization to accurately reconstruct large fossil specimens in 3D.

The detailed description of the three-dimensional skull and braincase presented herein also provides information that can be used in phylogenetic studies. Although incomplete, the skull and braincase preserve various elements that have not previously been reported or described in any specimen of Protoichthyosaurus and therefore it provides more information about this taxon so that its phylogenetic position can be explored in more detail. Furthermore, our study has found additional characters that may lend further support for the distinction of Protoichthyosaurus from its sister taxon Ichthyosaurus, such as the morphology of the pterygoid and anteroventral surface of the parietal, which differ from that described for Ichthyosaurus (McGowan, 1973). However, considering that only a couple of specimens preserve these elements, it is possible that the differences may be the result of individual variation; more three-dimensional specimens of both taxa are needed to test and clarify these findings.

Based on a unique combination of characters, we identify the studied specimen as P. prostaxalis. With a skull nearly twice as long as any previously described specimen of P. prostaxalis, this specimen greatly increases the known size range of this genus. Compared with known, contemporaneous Sinemurian ichthyosaurs, the estimated size suggests it was larger than all species of Ichthyosaurus (Lomax & Sachs, 2017), and comparable with the largest known specimens of Leptonectes tenuirostris (McGowan, 1996a), but smaller than L. solei (McGowan, 1993), Excalibosaurus costini (McGowan, 2003) and Temnodontosaurus platyodon (McGowan, 1996b). Thus, our study also provides new information on ichthyosaur diversity and potential ecology in the Early Jurassic of the UK.

Supplemental Information

Supplemental Information 1 3D PDF of segmented CT scans of the reassembled skull of Protoichthyosaurus prostaxalis (BMT 1955.G35.1).

Download the PDF file and click once on the skull to activate. Left-click to rotate the model; right-click to zoom in or out; and hold both buttons to pan. Check or uncheck boxes in the model tree in the upper left corner of the viewer to display or hide individual parts.

Click here for additional data file.

Supplemental Information 2 Transformation matrices for the 3D digital reconstruction of Protoichthyosaurus prostaxalis (BMT 1955.G35.1) from original CT data.

Click here for additional data file.

Supplemental Information 3 3D PDF of the reconstructed skull of Protoichthyosaurus prostaxalis (BMT 1955.G35.1).

Click here for additional data file.

Firstly, we would like to thank Luanne Meehitiya (formerly of TSM) who first discussed BMT 1955.G35.1 with NRL and DRL, and provided access to study the specimen. Thanks to Marta Fernández, Benjamin Moon and Alfio Chiarenza for their thorough and constructive reviews of this manuscript, which were greatly appreciated. We also thank Lukas Large (TSM) for assistance and the Birmingham Museums Trust for the photos reproduced in Figs. 1B and 1C. Judy Massare and Valentin Fischer are acknowledged for helpful comments on the morphology of the skull. Jon Radley provided information on the geology of the site and records relating to the excavation. David Ward identified the shark tooth and provided stratigraphic information about the species. Michael Howarth identified the ammonite associated with the postcranial skeleton and the ammonites recently collected by NRL from the original site. The current landowner R. E Morley and the tenant farmer Robert Heath gave permission for large holes to be dug at the site and Malcolm Bryan and Sally Bryan (son and wife of the original finder of the specimen, Michael Bryan) and Clive Jeffries assisted with the fieldwork. Robert Asher and Colin Shaw (University of Cambridge) provided access to microCT-scanning facilities; Renate Weller (Royal Veterinary College) carried out CT-scanning of the full skull. Technical support for Avizo was provided by Alejandra Sánchez-Eróstegui and Jean Luc-Garnier (Thermo Fisher Scientific, Waltham, MA, USA). Steven Dey (ThinkSee3D) mirrored microCT data and 3D printed the missing bones on the right side of the skull.

Institutional Abbreviations

BMT Birmingham Museums Trust (encompasses BMAG, Birmingham Museum and Art Gallery and TSM, Thinktank, Birmingham Science Museum), UK

BRLSI Bath Royal Literary and Scientific Institution, Bath, UK

BU Lapworth Museum of Geology, University of Birmingham, UK

LEICT Leicester Arts and Museums Service, New Walk Museum and Art Gallery, Leicester, UK

NHMUK Natural History Museum, London, UK

SOMAG (formerly AGC) Alfred Gillett Collection, cared for by the Alfred Gillett Trust (C & J Clark Ltd.), Street, Somerset, UK

UNM University of Nottingham Museum, UK.

Additional Information and Declarations

Competing Interests

Author Contributions

Data Availability

The authors declare that they have no competing interests.

Dean R. Lomax conceived and designed the experiments, performed the experiments, analysed the data, contributed reagents/materials/analysis tools, prepared figures and/or tables, authored or reviewed drafts of the paper, approved the final draft.

Laura B. Porro conceived and designed the experiments, performed the experiments, analysed the data, contributed reagents/materials/analysis tools, prepared figures and/or tables, authored or reviewed drafts of the paper, approved the final draft.

Nigel R. Larkin conceived and designed the experiments, performed the experiments, analysed the data, contributed reagents/materials/analysis tools, approved the final draft.

The following information was supplied regarding data availability:

Original CT data of the full skull is available at MorphoSource, here: https://www.morphosource.org/Detail/MediaDetail/Show/media_id/29156. Raw microCT data of individual braincase elements is not available as these data were lost; however, STLs and 3D PDFs of these elements are available (see here: https://www.morphosource.org/Detail/MediaDetail/Show/media_id/33743) and all elements that were individually scanned were also scanned with the full skull.

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
