# Peer review of "Descriptive anatomy of the largest known specimen of Protoichthyosaurus prostaxalis (Reptilia: Ichthyosauria) including computed tomography and digital reconstruction of a three-dimensional skull"

_PeerJ, doi:10.7717/peerj.6112_

## Round 0.1 · original submission · Minor Revisions

Dear authors,

I am sorry for the delay in this decision. I have accepted the decision of 'minor revision' from the three reviewers. However, due to the number of comments, and those from reviewer three, this should be considered between 'minor revisions' and 'major revisions'.

I have some additional comments that the authors should address prior to resubmission (in addition to those made by the reviewers):

1. Authority and date should be provided for each species-level taxon at first mention. Please ensure that the nominal authority is also included in the reference list.
2. When discussing other taxa in a comparative sense (such as in the description) can the authors give the full binomial instead of just the generic name.

Please note that PeerJ does not do a full linguistic check. That is the responsibility of the authors.

Once again, thank you for submitting your manuscript to PeerJ and I look forward to receiving your revised submission.

·

Basic reporting

The study is clear, and the English used in the text is professional and very well written.
The references reported cover sufficiently the areas mentioned in the text, but I have provided a suggestion for an addition related to some background info on the diversity and extinction of the group mentioned in the paper.
A rich series of figures and tabled data are shared with the reader providing an important source for professionals interested in this group.
The structure of the article does conform to the ‘standard sections’ indicated by the Instructions for Authors PeerJ’s regulation.
The figures present in the paper are contextualized throughout the text, of excellent quality and properly labelled.
Data figured and described in the text are also provided via a 3D supplementary material file.

Experimental design

As this study represent a redescription with a new taxonomic interpretation for an historical specimen of Mesozoic marine reptile, it represents an original primary research study.
This submission defines background and aim of the study clearly, honestly and without overselling it.
The investigation performed is rigorous and combines classic biostratigraphical, anatomical observations of fossil material, and cutting edge medical imaging techniques, reporting appropriately the machinery utilized herein and information useful to replicate the study or the methodology to whomever might be interested in doing so.

Validity of the findings

See general comment below. As the study represent a morphological, taxonomic and anatomical description, it is well self-contained and doesn’t report any speculation throughout.

Additional comments

General comments for the review of Lomax et al. “Descriptive anatomy of the largest known specimen of Protoichthyosaurus prostaxalis (Reptilia: Ichthyosauria) including computed tomography and digital reconstruction of a three-dimensional skull”.

This is an excellent, well-detailed study describing a historically valuable specimen housed at the Thinktank, Birmingham Science, UK. Although the specimen was originally identified as belonging to Ichthyosaurus communis (one of the most common attribution for Jurassic ichthyosaur specimens found in Great Britain), Lomax and coauthors performed a great deconstructionist study to investigate the identity of this specimen, making it scientifically valuable and available to the community.
Lomax and colleagues traced down the original locality, constraining the stratigraphic contest and age of the specimen, described the anatomy in detail with the aid of non-invasive 3D imaging reconstruction, and reviewed its original taxonomic attribution.
Considering that many Early Jurassic ichthyosaur specimens are often preserved in 2D, this specimen represents a rare opportunity to cast some additional light into the inner soft anatomy, particularly of the skull. In this paper, the authors not only describe the osteology of this specimen, but also restore and detail the inner soft anatomy of the brain and other related structures, providing a much useful example for the anatomy of this taxon to ichthyosaur workers.
The description they present and the taxonomic attribution are convincing.
The study is very well written and acceptable as it is, but I have flagged a few comments in the review pdf (see attached file) that may improve an already worthy to be published study.
In particular:

Line 40: I would add a citation here: I’d suggest Fisher et al. 2016.

Line 85: Add size information between brackets here: although the authors mention this information further on close to the end of the paper, it is a little bit frustrating mentioning it at first and then not giving away the information. The reader will appreciate more the immediacy, which will also be found more easily by interested readers.

Line 159: remove ‘and’ and replace it with ‘like’. I know perfectly what the authors mean here, but some relatively more unaware readers may interpret this as considering dinosaurs non-tetrapods (which we know not being the case).
In conclusion, I congratulate with the authors of this contribution and recommend the study for publication in PeerJ.

Reference

Fischer V, Bardet N, Benson RBJ, Arkhangelsky MS, Friedman M. 2016 Extinction of fish-shaped marine reptiles associated with reduced evolutionary rates and global environmental volatility. Nature Communications. 7, 10825. (doi:10.1038/ncomms10825)

Alfio Alessandro Chiarenza
Imperial College London

·

Basic reporting

Lomax and co-authors carried out CT scanning of a large Lower Jurassic ichthyosaur and
provide very useful and needed information on the skull anatomy of this peculiar reptiles.
CT scanning provides crucial information on the anatomy of extinct forms and it is much needed in Ichthyosauria literature (up to now only 2 contributions including this). The authors explore, by means of medical computed tomography scanner, a large skull of a Mesozoic marine reptile and provide three-dimensional digital reconstruction of a large skull of an ichthyosaur. In this way they bring a reliable background for the correct interpretation of skull anatomy and highlighted the utility of medical imaging techniques as powerful tool for exploring and studying even large specimens.

Certainly is the kind of manuscript that it will become a mandatory reference for all contributions focused on systematic and diversity of Jurassic ichthyosaurs. Reliable anatomical information is the only solid framework for hosting higher levels of inferences (such as phylogenetic and palaeoecological hypotheses) so I hope that this contribution will encourage others to go on exploring the anatomy of the ichthyosaurs from other ages and also for other marine reptiles.
I recommended its publication after minor revision

Experimental design

No comment.

Validity of the findings

No comment.

Additional comments

I congratulate the authors for this contribution. The authors did an outstanding job with the CT scanning, description and reconstruction of a large 3D preserved skull of Protoichthyosaurs. The detailed information and figures provided herein will be useful and a mandatory reference for all contributions dealing with the systematic and phylogeny of Jurassic ichthyosaurs.

The paper is self-contained, well written, it has a logical flow, and it can be read without ambiguities throughout. As I am not a native speaker I don´t feel confident to review the English grammar and/or provide further comments on the English.
I made only few suggestions and comments (see annotated PDF for more particular issues).
In summary, I think it is a valuable and needed contribution to our understanding of Lower Jurassic ichthyosaurs and it provides a solid basis to house higher levels of hypotheses on the evolution of the whole lineage. I recommend its publication after minor revisions

Minor comments and suggestions:

I think that some topics are worthy to be remark and/or certain questions that could be addressed such as:

What is the significance of the new anatomical information in a phylogenetic framework? The new information permits to complete the character scoring of the taxon? This will be very useful to determine the phylogenetic location of Protoichthyosaurus.

Which is the interpretation of the peculiar vascular system of the snout? Is it similar or not to the one described by Foffa et al. 2014. in pliosaurs?

It could also be worthy if the authors can provide more details and/or reference on how the total body length is estimated.

Several localities along the United Kingdom (UK) are mentioned throughout the text and there is no map. English fossiliferous localities (particularly southern ones) are worldwide famous so some readers could be family with most of them but not with all of them. The focus of the contribution makes not necessary to include a map but, if localities are mentioned it would be necessary to explain the significance and/or meaning of these information. Maybe the locality references must not only restrict to actual political divisions of the territories but also to geological formations outcropping in more general areas like Southern, center, northern… England or Scotland, etc.

Maybe the authors could include a reference: “Sollas (1916) The skull of Ichthyosaurus, Studies in Serial Sections…”. This contribution is, as far as I know, the first study (using different technic, of course) that provides serial sections of the skull of an ichthyosaur.

Finally, I have a minor comments on the taxonomical identification. From the two characters used by the authors to separate the two species of Protoichthyosaurus (maxilla and postorbital morphologies) I think that in the case of the BMT 1955.G35.1 only the maxilla can be used as the postorbital is lost. In any case, if the authors think that the morphology of the maxilla is not enough the specimen could be referred as Protoichthyosaurus sp.

·

Basic reporting

The authors present a new description of an ichthyosaur specimen and assign it to the taxon _Protoichthyosaurus prostaxalis._ This is well-presented and provides a good description of this material based on CT scans that is of suitable length for the report.

I have found some minor issues that are included in this document and the accompanying marked-up PDF, and suggest **moderate revision** of the work before publication. I also think that some further comparison and discussion of the anatomy of this taxon is necessary, particularly in the context of infrequently-preserved palatal elements. I am happy to be identified and contacted further by the authors to clarify any of my comments.

The English throughout is correct and used, however, in some case the lack of proper anatomical orientation nomenclature makes the description confusing. I have marked such instances in the accompanying PDF.

The introduction and literature review are suitable and reference relevant articles. Structure is appropriate.

The figures given are not as high-quality or structured as well as I think they should be. In particular, figs 2 and 4, which are the main references for the description of the anterior skull and jaw, are too small for purpose: these _are_ the main images to point features that are mentioned in the text and thus should be much larger. Even at full screen width on my laptop, they are too small and low-resolution for me to certainly see many features that you mention. This may partially be due to the resolution of the scans, but given the JPEG(?) artefacts I do not think that is the whole case. The 3D PDF provides this information, but without the essential labelling and scaling that the static figures are there to give. Additionally, I find the colour of the premaxillae to be difficult to see on a white background; this should be changed. Perhaps also consider the colouring and its visibility for colour blind people.

The RAW data of the complete skull scans in medical CT has been included and matches the segmented images (figs 2, 4) and 3D PDF as expected, but I did not get the scan data for the μCT scans for the braincase elements. These must be made available also.

Experimental design

Research is original and well-defined, relevant, and meaningful: the new description as some detail to the knowledge of ichthyosaur cranial anatomy.

The methods used are well explained, but some more details of the CT scan settings are required for complete reproducibility: particularly exposure length, filtering used. More details are needed on how and why a similar voxel size was used for the braincase elements.

Validity of the findings

The results and discussion are well supported, but could benefit from additional comparision with other know material; this is almost entirely lacking apart from the justification for placing in _Protoichthyosaurus prostaxalis._ In particular, fuller comparison against the limited palatal and endocranial material of other ichthyosaurs will be useful as a marker of ichthyosaur evolution, and morphological variation. Examples where some the palate is known include _Ichthyosaurus, Ophthalmosaurus, Platypterygius, Stenopterygius, Sveltonectes_ [McGowan 1973; Moon 2016; Baur 1895; Kear 2005; Fischer2011], which would all make pertinent comparative material.

Additional comments

# Additional considerations #

A few particular language considerations:

* Use anatomical orientations where required: e.g. 'lateral' or 'medial' (as relevant) over 'right' or 'left' (l 261); 'laterally' rather than 'outwards' (l 365).
* Ensure that directions 'left' or 'right', where used, are used correctly: in some instances it was unclear whether you were referring to left or right or paired elements, or to the left or right side of an element (e.g. description of the premaxilla \[l 261\]). More often these should be replaced with 'medial' and 'lateral'.

Several elements are described or mentioned as being present, but not given a formal description:

* Right nasal (l 299): mentioned as present but not figured.
* Postfrontal (l 323): mentioned as present and partially complete, but neither described nor figured.
* Supratemporal (l 425): additional figures of the lateral margin to show the additional facets should be included.

These elements should be figured to show they are difficult to describe, or should be included more fully.

## Maxilla ##

Does the maxilla show evidence of the external narial margin on its dorsal surface? This is one of the features of _P. prostaxalis_ described by @Lomax2018, as it would be useful to compare.

## Jugal ##

The left side, as preserved, shows this it extends anterior to the orbital margin, as marked by the prefrontal, but does not appear t be complete. Additionally, the facets on the lacrimal and maxilla appear to extend anterior to this. Is the extent of the left jugal going to go extensively anterior to the anterior orbital margin or minimal?

## Pterygoid ##

The comparisons between BMT 1955.G35.1, _Sveltonectes,_ and _Ophthalmosaurus_ (l 390) may need more justification: what are the similarities/differences? bearing in mind that the pterygoids of both this specimen and _Sveltonectes_ are incomplete compared to _Ophthalmosaurus._ The general form seems to be similar in the shape of the quadrate ramus, while only the posterior portion of the palatal ramus is preserved in BMT 1955.G35.1.

I can't see the dorsal ramus of the pterygoid clearly in either the figures or the 3D PDF. Can this be pointed out somewhere? Also, there is no description of the contact laterally with the quadrate: how extensive? how close? does it meet the stapes (as in _Ophthalmosaurus_ and _Leninia_ [Fischer 2014, Moon 2016])? These details, or mention that it is not describe-able.

## Quadrate ##

The groove dividing the jaw articulation into two sections is common in ichthyosaurs, so I don't know why you have picked out only _Acamptonectes._ Other mentions include: _Ichthyosaurus_ [McGowan 1973], _Ophthalmosaurus_ [Moon 2016], _Platypterygius_ [Kear 2005].

# References #

Baur, G. 1895. “Die Palatingegend der Ichthyosauria.” Anatomischer Anzeiger 10 (15): 456–59.

Fischer, Valentin, Maxim S Arkhangelsky, Gleb N Uspensky, Ilya M Stenshin, and Pascal Godefroit. 2014. “A New Lower Cretaceous Ichthyosaur from Russia Reveals Skull Shape Conservatism Within Ophthalmosaurinae.” Geological Magazine 151 (01): 60–70. doi:10.1017/S0016756812000994.

Fischer, Valentin, Edwige Masure, Maxim S Arkhangelsky, and Pascal Godefroit. 2011. “A New Barremian (Early Cretaceous) Ichthyosaur from Western Russia.” Journal of Vertebrate Paleontology 31 (5): 1010–25. doi:10.1080/02724634.2011.595464.

Kear, Benjamin P. 2005. “Cranial Morphology of Platypterygius Longmani Wade, 1990 (Reptilia: Ichthyosauria) from the Lower Cretaceous of Australia.” Zoological Journal of the Linnean Society 145 (November): 583–622. doi:10.1111/j.1096-3642.2005.00199.x.

Lomax, Dean R, and Judy A Massare. 2018. “A Second Specimen of Protoichthyosaurus Applebyi (Reptilia: Ichthyosauria) and Additional Information on the Genus and Species.” Paludicola 11 (4): 164–78.

McGowan, Christopher. 1973. “The Cranial Morphology of the Lower Liassic Latipinnate Ichthyosaurs of England.” Bulletin of the British Museum (Natural History), Geology 24 (1): 1–109.

Moon, Benjamin Christopher, and Angela M Kirton. 2016. “Ichthyosaurs of the British Middle and Upper Jurassic. Part 1, Ophthalmosaurus.” Monograph of the Palaeontographical Society 170 (647): 1–84. doi:10.1080/02693445.2016.11963958.

---

## Round 0.2 · accepted · Accept

Dear authors,

Many thanks for your revised manuscript. After reading it, I have accepted it for publication in PeerJ.

Once again, thank you for submitting your manuscript to PeerJ and I hope you will use us again as your publication venue.

If we need to clarify any details required to move the manuscript forward, then our production staff will get in touch with you. Otherwise, a proof will be forthcoming shortly for your review.

Congratulations and thank you for your submission.

# ·

Basic reporting

I have no further comments to those expressed in my previous review. I agree with the answers and / or refutations to my former comments. I congratulate the authors for this interesting contribution that will become a mandatory reference for future research on Jurassic ichthyosaurs. I hope that this contribution encourages others to deepen the anatomical exploration of these great marine reptiles.

Experimental design

no comment

Validity of the findings

no comment

Additional comments

I have no further comments to those expressed in my previous review. I agree with the answers and / or refutations to my previous comments. I congratulate the authors for this interesting contribution that will become a mandatory reference for future research on Jurassic ichthyosaurs. I hope that this contribution encourages others to deepen the anatomical exploration of these great marine reptiles.

·

Basic reporting

I find this to be appropriate and thorough throughout. My comments from the first version have been addressed suitably.

Experimental design

As the original scan data has been lost, I think this should be mentioned either in-text or in the supplement to explain why it is not included; this will (hopefully) prevent the authors being asked in future why it wasn't included too.

Validity of the findings

No comment

Additional comments

Thank you for your replies to the modifications I requested.

A lot of my comments concerned presentation of figures, so I wish to justify myself: my view is that as much of what is discussed in text should be included in the figures alongside as possible, particularly described features, and even if a separate 3D or other data set is available. I realise this is not always possible, but have tried to match these standards. This is why I requested larger, higher resolution images, and more images of elements that did not show the described features. It is possible in Avizo to export images larger than the window size using the 'Render tiles' option, and avoiding JPEG compression where possible by using TIFF of vector files is a good option. The supratemporal in Fig 6A–C does not clearly show all the facets described.

However, given the presence of the 3D PDF I am happy with the quality and completeness of the figures presented by the authors if the editors are also.